

# The preclinical pharmacological study of a novel intravenous anesthetic, ET-26 hydrochloride, in aged rats

Pan Chang[1,*], YongWei Su[1,*], DeYing Gong[2,3], Yi Kang[2,3], Jin Liu[1,2,3], YuJun Zhang[1,2,3] and Wen-sheng Zhang[1,2,3]

[1] Department of Anesthesiology, West China Hospital, Sichuan University, Chengdu, China
[2] Laboratory of Anesthesia and Critical Care Medicine, Translational Neuroscience Center, West China Hospital, Sichuan University, Chengdu, China
[3] National-Local Joint Engineering Research Center of Translational Medicine of Anesthesiology, West China Hospital, Sichuan University, Chengdu, China
[*] These authors contributed equally to this work.

Corresponding authors
YuJun Zhang, zhangyujun@scu.edu.cn
Wen-sheng Zhang, zhang_ws@scu.edu.cn

## ABSTRACT

**Background**. ET-26 hydrochloride (ET-26HCl) is a novel analogue of etomidate approved for clinical trials. However, all results from recent studies were accomplished in young adult animals. The objective of this study was to evaluate the efficacy and safety of ET-26HCl in aged rats.

**Methods**. Aged Sprague-Dawley rats were randomly divided into three groups (three males and three females in each group) were given dose of two-fold of median effective dose ($ED_{50}$) of ET-26HCl, etomidate and propofol: the measurements of loss of the righting reflex (LORR) and cardiovascular and respiratory function after injection at the two-fold dose of the median effective dose were used for evaluation of effectiveness and safety, and the modified adrenocorticotropic hormone-stimulation experiment was used to evaluate the inhibition effect of the drugs on the synthesis of adrenal cortical hormones.

**Results**. There was no significant difference in the onset time among propofol, etomidate and ET-26HCl. The duration of propofol ($850.5 \pm 77.4$ s) was significantly longer than that caused by etomidate ($489.8 \pm 77.0$ s, $p = 0.007$) and ET-26HCl ($347.3 \pm 49.0$ s, $p = 0.0004$). No significant difference was observed in the time to stand and normal activity among drugs. A total of 66.7% of rats in the ET-26HCl group were evaluated to have mild hematuria. Then, etomidate and ET-26HCl had a milder blood pressure inhibition effect than propofol. Apnea was observed in all rats administered propofol and the duration for this side effect was $45.0 \pm 9.0$ s. For etomidate and ET-26HCl, no apnea was observed. No other clinical signs of side-effect were observed, and no rats died. No significant difference was observed in corticosterone concentrations between ET-26HCl and solvent group. However, rats administered etomidate had lower corticosterone concentrations than those administered ET-26HCl at 15, 30, and 60 min.

**Conclusions**. Our results indicate ET-26HCl in aged rats is an effective sedative-hypnotic with stable myocardial and respiratory performance and also have mild adrenocortical suppression. Thus, these findings increase the potential for the clinical use of ET-26HCl in the elderly population.

## INTRODUCTION

The aging global population is a universal social situation. The number of aged people reached 600 million in 2000 and is predicted to reach 2.1 billion in 2050 (*McNicoll, 2002*). Coexisting diseases, especially in the cardiovascular system, are more prevalent in the elderly than in the young adult population. A commonly used intravenous anesthetic, propofol, has extensive effects on the cardiovascular and respiratory system (*Sahinovic, Struys & Absalom, 2018*). The most prominent side-effect of propofol is cardiovascular depression, especially for blood pressure (BP) reduction (*Ebert, 2005*; *Hug Jr et al., 1993*). Additionally, propofol decreases the sensitivity of central chemoreceptors, reduces tidal volume, and leads to apnea (*Jonsson, Lindahl & Eriksson, 2005*; *Nieuwenhuijs et al., 2000*; *Nieuwenhuijs et al., 2001*; *Yamakage et al., 1999*). Due to its excellent cardiovascular stability and mild respiratory depression, etomidate is the most widely used intravenous anesthetic for elderly patients (*Giese et al., 1985*; *Tomlin et al., 1998*; *Ye, Xiao & Zhu, 2017*). However, previous studies have indicated that etomidate may even reduce the survival rate in the perioperative period *via* the inhibition of the synthesis of adrenal cortical hormones (*Chan, Mitchell & Shorr, 2012*; *Fellows, Byrne & Allison, 1983*; *Forman, 2011*; *Kamp & Kress, 2007*). Despite this severe adverse effect, etomidate is still the current optimal option for the induction of anesthesia in elderly patients (*Bovill, 2006*).

Etomidate is an imidazole-based agonist of the $\gamma$-aminobutyric acid type A receptor used for the induction of general anesthesia and sedation (*Valk & Struys, 2021*). Previous studies demonstrated that etomidate-inhibited adrenal steroidogenesis was related to its ester side chain (*Atucha et al., 2009*). Meanwhile, the primary metabolic production of etomidate, etomidate acid, had a lower inhibition of the synthesis of adrenal cortical hormones (*Zolle et al., 2008*). Thus, a new etomidate analogue was designed and synthesized to modify the ester side chain or metabolize to etomidate acid rapidly for reducing the adrenocortical suppression effect of etomidate. ET-26 hydrochloride (ET-26HCl; patent number, US9969695B2; Fig. 1) was screened as a novel intravenous anesthetic with not lonely similar extent of cardiovascular stability and respiratory depression comparing to etomidate, but also milder adrenocortical inhibition (*Wang et al., 2017c*). ET-26HCl was approved by the China National Medical Products Administration (NMPA) for clinical trials in 2019.

In non-clinical pharmacological studies, ET-26HCl produces a desirable sedative-hypnotic effect with stable myocardial performance and mild adrenocortical suppression in young animals (*Liu et al., 2018*; *Wang et al., 2017c*; *Yang et al., 2017*). In a rat uncontrolled hemorrhagic shock model, ET-26HCl shows similar hemodynamic stability to etomidate (*Wang et al., 2017a*). Furthermore, in a rat lipopolysaccharide-induced sepsis model, ET-26HCl induces a higher survival rate, a lower inflammatory reaction, and a reduced organ injury than etomidate (*Wang et al., 2017b*). These findings indicated that ET-26HCl

**Figure 1** Chemical structure of ET-26HCl.

can be an effectively sedative-hypnotic anesthetic with similarly excellent cardiovascular stability as etomidate and milder adrenocortical inhibition than etomidate. However, all results from recent non-clinical pharmacological studies were accomplished in young animals. Thus, there is currently no direct evidence that shows the efficacy and safety of ET-26HCl in the elderly population.

In this study, we compared the pharmacodynamic properties of ET-26HCl, etomidate, and propofol with the injection of the two-fold dose of the $ED_{50}$ for LORR in aged rats. All rats were observed closely for mortality and clinical symptoms of toxicity. We further evaluated the effect of these drugs on mean arterial pressure, heart rate, respiratory rate, and adrenocortical suppression in aged rats.

## MATERIALS AND METHODS

### Animals and housing

All animal procedures were performed with the approval of the Institutional Animal Care and Use Committee of West China Hospital, Sichuan University, Chengdu, China (2015015A). All animals were housed according to the Guide for the Care and Use of Laboratory Animals by the West China Hospital, Sichuan University. Sprague-Dawley SPF rats (age: 20 months, body weight: $680 \pm 23$ g for male and $508 \pm 59$ g for female after 5 days adaptation) were purchased from Dossy Biological Technology Co., Ltd. (Chengdu, China). Rats ($n = 5$) were housed in polypropylene cages with *ad libitum* food and water supply, controlled room temperature (22 °C to 26 °C) and humidity (40% to 60%), and under 12 h light-dark cycles (lights on at 7:00 a.m.). Prior to the experiments, animals were acclimated for a minimum of 5 days.

### Materials

ET-26HCl was synthesized by Humanwell Pharmaceutical Co., Ltd. (Yichang, China). The solution of ET-26HCl (10 mg/mL) was dissolved in 35% (*v/v*) propylene glycol and sterile injection water, and this specific solvent was prepared by Humanwell Pharmaceutical

Co., Ltd. Propofol (10 mg/mL) and etomidate (10 mg/mL) were purchased from AstraZeneca (Shanghai, China) and Nhwa Pharmaceutical Co., Ltd. (Jiangsu, China), respectively. Isoflurane was purchased from Yipin Pharmaceutical Co., Ltd. (Hebei, China). Dexamethasone sodium phosphate injection and normal saline were purchased from Tianyao Pharmaceutical Co., Ltd. (Hubei, China) and Qingshanlikang Pharmaceutical Co., Ltd. (Chengdu, China), respectively. Adrenocorticotropic hormone (ACTH) fragment 1-24 (human and rat) was purchased from Sigma-Aldrich (MO, USA) and dissolved with dimethyl sulfoxide (20 $\mu$g/mL).

Corticosterone and corticosterone-d8 were purchased from the Dr. Ehrenstorfer GmbH (Augsburg, Germany) and $C/D/N$ Isotopes Inc. (Quebec, Canada), respectively. Acetonitrile, formic aid, and dimethyl sulfoxide were purchased from Sigma-Aldrich. Ethyl acetate, ethyl alcohol and methanol were obtained from Thermo Fisher Scientific (NJ, USA). All reagents are HPLC grade solvents. Ultrapure water was generated by the Milli-Q$^®$ water purification system (Merck Millipore, Darmstadt, Germany).

## Experimental design

Eighteen aged Sprague-Dawley rats were randomly divided into three groups (three males and three females in each group). According to a previous study (*Wang et al., 2017c*), the $ED_{50}$ values for LORR in rats are 2.35 mg/kg, 0.73 mg/kg, and 5.87 mg/kg for ET-26HCl, etomidate and propofol, respectively. Rats were injected intravenously with the two-fold dose of $ED_{50}$ for LORR *via* the tail vein in an injection speed of 0.1 mL/s.

## LORR measurement

Rats were separately measured for LORR in a transparent box with a thermostatically heated pad (37 °C) to avoid external stimuli. The hypnotic depth was scored continuously until the rat was fully awake according to the following criteria (*Salamone et al., 1996*; *Shimoyama et al., 1999*; *Wang et al., 2017c*): 0 = normal activity; 1 = decreased activity or remained in same position; 2 = could not balance on hind limb; 3 = ataxia; 4 = lay prone and no reaction to mild stimuli; 5 = LORR. Once the score achieved 4, the rat was gently changed to the supine position (recognized as LORR). The time to LORR, time to recovery from LORR, time to stand up, and time to normal activity represented the onset, duration, and recovery of sedative effect. All rats that recovered from LORR were observed closely for mortality and clinical symptoms of toxicity.

## Cardiovascular and respiratory function measurement

Rats were anesthetized with 2% isoflurane in oxygen using a mask that was placed in the supine position on a thermostatically heated pad (37 °C). The tail vein was cannulated with a Terumo$^®$ intravenous catheter (24G; Tokyo, Japan) for drug administration. Then, the left femoral artery was isolated and cannulated for monitoring BP. A three-lead electrocardiogram (ECG) was monitored by the subcutaneous placement of an electrode at the left upper limb, right upper limb, and left lower limb. Thoracic movement was measured by a tension sensor for respiratory rate (RR). The ECG, BP and RR were recorded by the BL-420 Biological Signal Acquisition and Analysis System (Techman, Chengdu, China). Rats were acclimated for 20 min prior to the initiation of dosing. In the adaptive phase,

the parameters were collected every 4 min and averaged as the baseline. Then, rats were injected with the two-fold dose of the $ED_{50}$ for LORR as described in 2.3. The biological parameters were used to compare the degree of cardiovascular and respiratory system inhibition by drugs at the following time points: 0.5, 1 min, and every 1 min to 15 min. The mean arterial pressure (MAP) was calculated according to the equation below:

$$\text{Mean arterial pressure} = \left(\text{systolic pressure} + 2 \times \text{diastolic pressure}\right)/3.$$

## Serum corticosterone measurement
### Modified adrenocorticotropic hormone (ACTH)-stimulation experiment
The modified ACTH-stimulation experiment was used to evaluate the inhibition effect of the drugs on the synthesis of adrenal cortical hormones (*Cotten et al., 2009*; *Pejo et al., 2012*; *Wang et al., 2017c*). Drug administration and blood sample collection were completed *via* the tail vein with a cannulated catheter (Terumo®, 24G; Tokyo, Japan). Dexamethasone (0.5 mg/kg) was injected to inhibit corticosterone synthesis. Two hours after the administration of dexamethasone, ET-26HCl and etomidate were administrated at 4.70 mg/kg and 1.46 mg/kg (two-fold dose of the $ED_{50}$ for LORR), respectively. In the control group, 35% (*v/v*) propylene glycol was injected at 0.47 mL/kg (the same volume of ET-26HCl). Fifteen minutes later, ACTH (25 μg/kg) was administered to accelerate corticosterone production. The concentration of corticosterone was compared at the baseline, 2 h after dexamethasone injection, 15 min, 30 min, and 60 min after ACTH administration. Blood samples (0.2 mL each) were centrifuged at 870 g for 10 min. Serum was collected after clotting within 1 h and was stored at −20 °C.

### Sample preparation and liquid chromatography-mass spectrometry (LC-MS/MS) analysis
For corticosterone measurement, 50 μL of sample was added into a polypropylene tube and 150 μL of acetonitrile solution that contained corticosterone-d8 (20 ng/mL; internal standard, IS) was added to deproteinize the serum. After centrifuging (10,300 g, 10 min, 4 °C), the supernatant was analyzed *via* LC-MS/MS (Agilent Technologies 6460 triple quadrupole mass spectrometer with electrospray ionization source). Chromatographic separation was carried out using Waters symmetry C18 column (3 mm × 100 mm, 3.5 μm) at 30 °C. The mobile phase consisted of 0.1% formic acid (A) and acetonitrile (B) in a gradient elution: 0 min (A at 60%), 1 min to 1.4 min (A at 40%), and 4.5 min to 4.6 min (A at 10%) at a flow rate of 0.3 mL/min. Mass spectrometry was in positive ionization mode: sheath gas flow rate, 11.0 L/min; sheath gas heater temperature, 300 °C; nebulizer pressure, 45 psi; capillary voltage, 3,500 V. MassHunter software was used to analyze data (B.04.00 Build 4.0.479.0; Agilent Technologies, CA, USA).

## Data analysis
For pharmacodynamic experiments, data were assessed by one-way analysis of variance (ANOVA) followed by the Bonferroni post-hoc test ($\alpha = 0.025$). For comparison of cardiovascular function and adrenal cortical hormones inhibition, data were assessed by two-way ANOVA followed by the Bonferroni post-hoc test ($\alpha = 0.025$). For respiratory

function measurement, data were assessed by Fisher's exact test ($\alpha = 0.025$). Data are reported as mean $\pm$ SEM, and the level of statistical significance was set at $P < 0.05$. Data were analyzed using Statistical Package for Social Sciences version 23 (IL, USA).

## RESULTS

### Pharmacodynamic experiments

After the injection of the drugs at the two-fold dose of $ED_{50}$ for LORR, there was no significant difference in onset time among those drugs (Fig. 2A, $p = 0.053$). However, the duration of the sedative-hypnotic effect (score of hypnotic depth $= 5$, Fig. 2B) for propofol ($850.5 \pm 77.4$ s) was significantly longer than that caused by etomidate ($489.8 \pm 77.0$ s, $p = 0.007$) and ET-26HCl ($347.3 \pm 49.0$ s, $p = 0.0004$). Regarding recovery from the state of sedation (Fig. 2C), the time required to stand after administration of propofol, etomidate, and ET-26HCl was $71.3 \pm 33.5$ s, $140.3 \pm 42.6$ s, and $129.0 \pm 22.3$ s, respectively. The time required to normal activity after the administration of propofol, etomidate, and ET-26HCl was $393.0 \pm 91.2$ s, $382.0 \pm 52.6$ s, and $378.0 \pm 47.8$ s, respectively. However, no significant difference was observed in the time to stand ($p = 0.330$) and normal activity ($p = 0.987$) among the drugs.

Regarding the clinical symptoms of toxicity, 66.7% (4 of 6) of rats in ET-26HCl group exhibited mild hematuria (abnormal urine color—red), which was considered to be induced by the solvent. All rats recovered from their symptoms of toxicity, no other clinical signs of side-effect were observed, and no rats died.

### Cardiovascular and respiratory function measurement

The advantages of etomidate are better cardiovascular stability and milder respiratory depression than propofol. To test cardiovascular and respiratory inhibition in aged rats, BP, ECG, and RR were continuously monitored to explore the effects of the drugs (Fig. 3). In each group, MAP decreased after injection (Fig. 3A). The maximum value of MAP after the administration of etomidate, ET-26HCl and propofol was decreased at $23.8 \pm 3.1$ mmHg, $30.9 \pm 8.1$ mmHg, and $44.2 \pm 6.7$ mmHg, respectively. In etomidate and ET-26HCl groups, the inhibition of BP was temporary and it tended to recovery without any treatment. However, etomidate ($p = 0.0001$) and ET-26HCl ($p = 0.003$) had a milder BP inhibition effect than propofol. After drug administration, the etomidate-, ET-26HCl- and propofol-induced decrease in heart rate were not significantly different (Fig. 3B, $p = 0.12$).

The respiratory depression of the aged rats was compared *via* the measurement of occurrence rate and duration of apnea (Table 1). Apnea was observed in all rats that were administered propofol (6 of 6, 100%) and the duration of this side effect was $45.0 \pm 9.0$ s. For etomidate and ET-26HCl, no apnea was observed ($p = 0.00016$).

### Serum corticosterone measurement

In the modified ACTH-stimulation experiment, the concentration of serum corticosterone in aged rats is presented in Fig. 4. The concentration of corticosterone was suppressed by dexamethasone to $37.4 \pm 13.6$ ng/mL in the ET-26HCl group, $86.7 \pm 32.7$ ng/mL

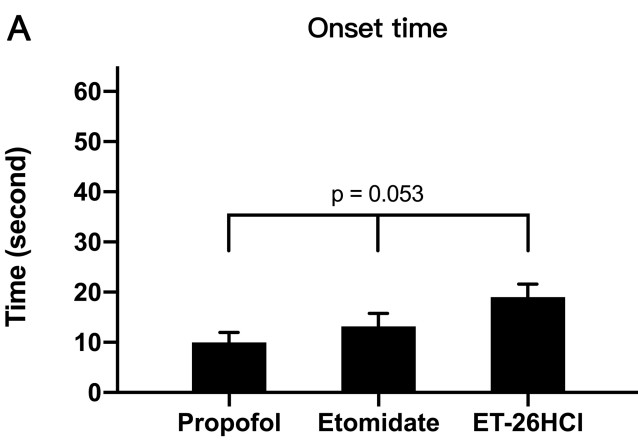

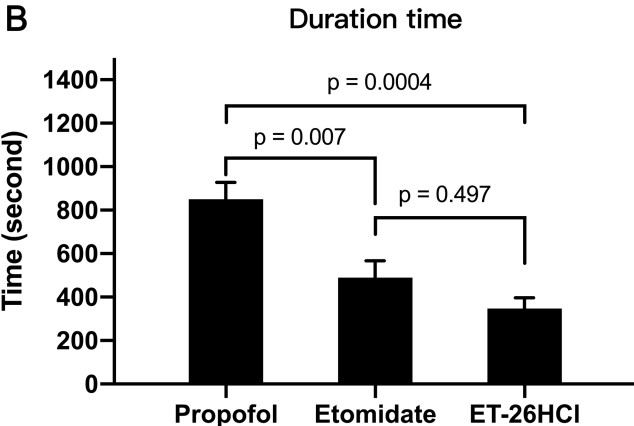

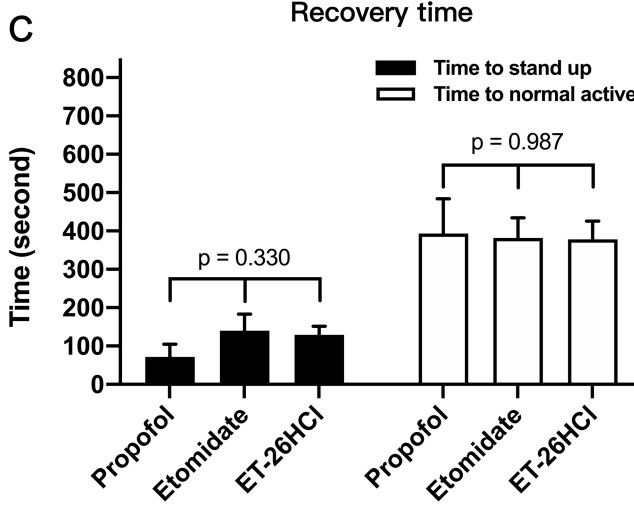

**Figure 2** The onset time (A), duration (B) and recovery time (C) after treatment with two-fold dose of the median effective doses for loss of the righting reflex in aged rats ($n = 6$ in each group).

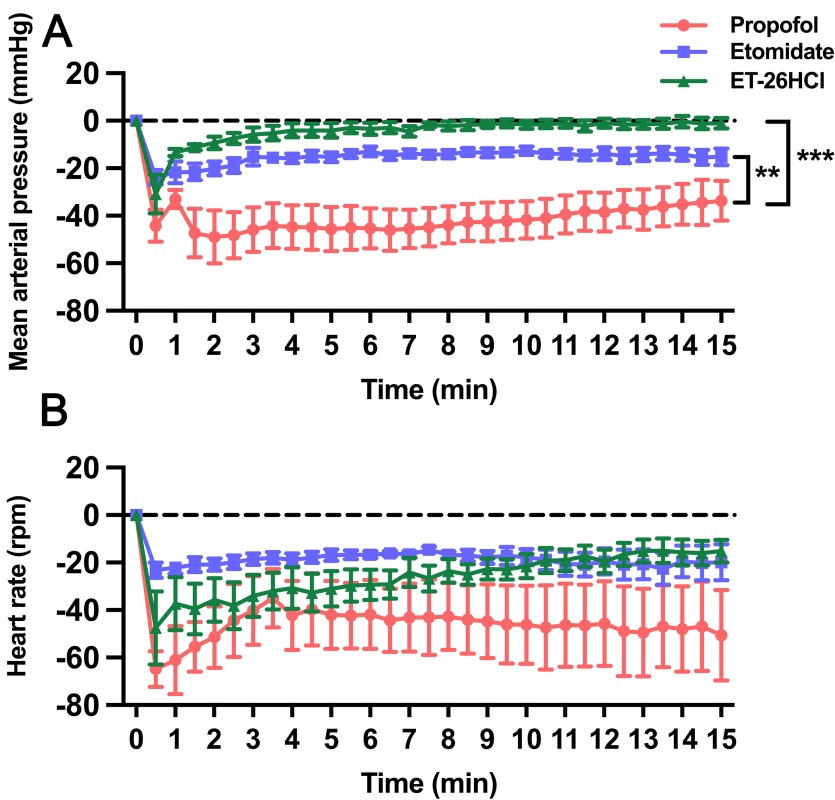

**Figure 3** The effect of ET-26HCl, etomidate, and propofol on mean blood pressure (A) and heart rate (B) after treatment with two-fold dose of the median effective doses for loss of the righting reflex in aged rats ($n = 6$ in each group). ** $p < 0.01$; *** $p < 0.001$.

**Table 1** The information of apnea after single administration of propofol, etomidate and ET-26HCl in aged rats (mean ± SD).

| Agent | $n$ | Number of aged rats with apneas | Apnea | | Duration of apnea (s) |
| --- | --- | --- | --- | --- | --- |
| | | | Onset time (s) | Offset time (s) | |
| Propofol | 6 | 6 | 7.5 ± 5.7 | 52.5 ± 9.2 | 45.0 ± 9.0 |
| Etomidate | 6 | 0[a] | 0 | 0 | 0 |
| ET-26HCl | 6 | 0[a] | 0 | 0 | 0 |

**Notes.**
[a] Compared with propofol group, $\chi^2 = 16.5$, $p = 0.00016$. Apnea in all rats was observed in propofol group; no apnea in etomidate and ET-26HCl groups.

in the etomidate group, and 39.8 ± 16.5 ng/mL in the solvent group, and there was no significant difference among groups ($p = 0.241$). No significant difference in the concentration of serum corticosterone was observed between ET-26HCl and solvent groups. However, rats administered etomidate had lower corticosterone concentrations than were administered ET-26HCl at 15, 30, and 60 min after the stimulation of ACTH ($p = 0.016$, $p = 0.004$, and $p = 0.002$, respectively). These results demonstrated that ET-26HCl had no obvious inhibitory effect on the synthesis of adrenal cortical hormones,

     

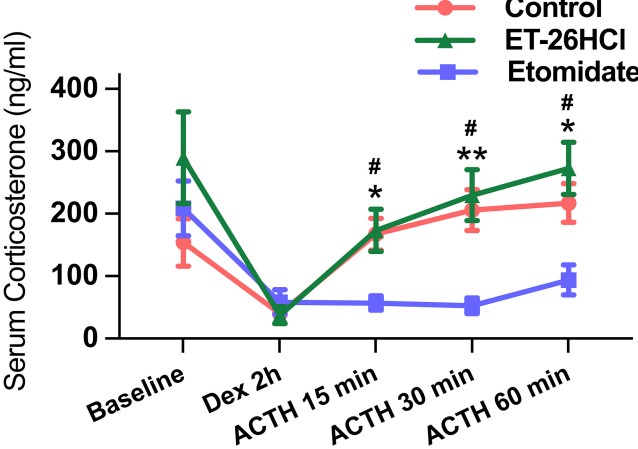

**Figure 4** The concentration of serum corticosterone stimulated by ACTH after treatment with two-fold dose of the median effective doses for loss of the righting reflex in aged rats ($n = 6$ in each group). Compared with etomidate, * $p < 0.05$ and ** $p < 0.01$, compared ET-26HCl with etomidate; # $p < 0.05$, compared control with etomidate.

whereas etomidate significantly decreased the synthesis of corticosterone within 1 h in aged rats.

## DISCUSSION

In this study, ET-26HCl was demonstrated to have similar advantages in aged rats as in young rats (*Liu et al., 2018*; *Wang et al., 2017a*; *Wang et al., 2017b*; *Wang et al., 2017c*; *Yang et al., 2017*), including an effective sedative-hypnotic effect, slight cardiovascular suppression, mild respiratory system and adrenocortical inhibition. Therefore, results of the current study can be translated to predict the potential advantages of ET-26HCl in the elderly population.

Aging induces pharmacokinetic and pharmacodynamic changes (*Butterworth, Mackey & Wasnick, 2018*). With decreases in muscle mass and increases in body fat, the total body water of elderly patients is decreased, especially in women. The required dose of drug for the same anesthetic effect is lower in elderly patients than that in young patients. In pharmacodynamic experiments, there is no significant difference between young rats (*Wang et al., 2017c*) and aged rats in the onset time, duration, time to stand, and time to normal activity after administration of the same dose of drug. Thus, as in young rats, ET-26HCl causes a shorter duration with no significant difference in recovery than propofol in aged rats.

In young rats, ET-26HCl and etomidate produce a milder reduction in MAP than propofol at equal hypnotic doses (*Wang et al., 2017c*). The good hemodynamic stability of ET-26HCl is maintained in aged rats. ET-26HCl reduced the blood pressure with lesser reduce degree and shorter duration than propofol. Meanwhile, propofol is more likely to cause apnea and hypotension in aged patients than in younger patients (*Butterworth, Mackey & Wasnick, 2018*). We further compared the depression effect on the respiratory

rate. Apnea was not observed in aged rats injected with ET-26HCl and etomidate. Then, we have demonstrated that ET-26HCl has minimal adrenocortical suppression than etomidate in aged rats, as has already been shown in young rats (*Wang et al., 2017c*). No significant difference in the concentration of serum corticosterone was observed between ET-26HCl and solvent groups. However, etomidate still inhibited the synthesis of adrenal cortical hormones in aged rats. All results indicated that ET-26HCl may be safer for elderly patients than etomidate and propofol.

However, in the ET-26HCl group, mild hematuria was observed in 66.7% of rats, which could have been induced by the solvent of 35% (*v/v*) propylene glycol. A previous study demonstrated that the hemolytic effect of propylene glycol can be eliminated *in vivo* by the preparation of no more than 30% (*v/v*) propylene glycol with saline or water (*Ruddick, 1972*). However, the red blood cells of rabbit are not affected after the intravenous injection of 50% propylene glycol in saline (*Brittain & D'Arcy, 1962*). Etomidate formulated in propylene glycol induces hemolysis after bolus injection in patients and healthy volunteers (*Doenicke et al., 1997*; *Doenicke et al., 1999*). In the single dose and 14-days repeated-dose toxicity study of ET-26HCl in rats (*Zhang et al., 2020b*) and beagle dogs (*Zhang et al., 2020a*), the occurrence rate and severity of hemolysis are not significantly different between the treated and control groups (35% propylene glycol). Therefore, we considered that this side-effect was related to the solvent, not ET-26HCl. Furthermore, formulations of etomidate in 35% propylene glycol and water, such as Amidate[TM] (Pfizer Inc) and Hypnomidate[®] (Janssen Pharmaceutica Ltd), have been used in hospitals. Thus, we believe that the toxicity of this solvent should be noted in clinical trials; however, this toxicity is likely not severe.

There are still some limitations in this study. The required dose of drug for the anesthetic effect is lower in elderly patients than that in young patients (*Butterworth, Mackey & Wasnick, 2018*). The dosages in this study were according to the previous study in young rats (*Wang et al., 2017c*), and we did not measure the $ED_{50}$ for LORR in aged rats. Besides, the telemetry method for cardiovascular and respiratory function measurement in unrestrained conscious animal, which was more conform to the real clinical situation without the interference of other drugs, such as 2% isoflurane in this study.

## CONCLUSIONS

In this study, we demonstrated, for the first time in aged rats, that ET-26HCl reserves the advantages of etomidate in aged rats, including a mild cardiovascular and respiratory system suppression than propofol. ET-26HCl further eliminate the major disadvantage of etomidate, adrenocortical inhibition. Meanwhile, ET-26HCl possesses the same sedative-hypnotic effect as etomidate. This key discovery can be used to advise the use of ET-26HCl, rather than etomidate and propofol, for the induction of general anesthesia in elderly patients. However, clinical trials that involve the use of elderly patients is needed before any conclusions can be drawn.

## ACKNOWLEDGEMENTS

We are grateful to LingHui Yang and Jing Wang from the Laboratory of Anesthesia and Critical Care Medicine (West China Hospital, Sichuan University) for their technical supports.

### Funding

This work was supported by a grant from National Science and Technology Major Project, Ministry of Science and Technology of the People's Republic of China (No. 2014ZX09101001), Beijing, China; the Science and Technology Department of Sichuan Province, China (No. 20YYJC3881, 2018sz0236); and the Science and Technology Department of Chengdu City, China (No. 2021-YF05-00855-SN). The funders had no role in study design, data collection and analysis, decision to publish, or preparation of the manuscript.

### Grant Disclosures

The following grant information was disclosed by the authors:
National Science and Technology Major Project, Ministry of Science and Technology of the People's Republic of China: 2014ZX09101001.
Science and Technology Department of Sichuan Province, China: 20YYJC3881, 2018sz0236.
Science and Technology Department of Chengdu City, China: 2021-YF05-00855-SN.

### Competing Interests

The authors declare there are no competing interests.

### Author Contributions

- Pan Chang conceived and designed the experiments, performed the experiments, analyzed the data, prepared figures and/or tables, and approved the final draft.
- YongWei Su conceived and designed the experiments, performed the experiments, analyzed the data, prepared figures and/or tables, and approved the final draft.
- DeYing Gong conceived and designed the experiments, performed the experiments, analyzed the data, prepared figures and/or tables, and approved the final draft.
- Yi Kang conceived and designed the experiments, performed the experiments, analyzed the data, prepared figures and/or tables, and approved the final draft.
- Jin Liu conceived and designed the experiments, authored or reviewed drafts of the article, and approved the final draft.
- YuJun Zhang conceived and designed the experiments, performed the experiments, analyzed the data, prepared figures and/or tables, authored or reviewed drafts of the article, and approved the final draft.
- Wen-sheng Zhang conceived and designed the experiments, analyzed the data, authored or reviewed drafts of the article, and approved the final draft.

## Patent Disclosures

The following patent dependencies were disclosed by the authors:

   N-substituted imidazole carboxylic ester chiral compound containing an ether side chain, its preparation and application, US9969695B2.

## Data Availability

   The raw measurements are available in the Supplemental File.

## Supplemental Information

Supplemental information for this article can be found online at http://dx.doi.org/10.7717/peerj.13995#supplemental-information.

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
