# Peer review of "The preclinical pharmacological study of a novel intravenous anesthetic, ET-26 hydrochloride, in aged rats"

_PeerJ, doi:10.7717/peerj.13995_

## Round 0.1 · original submission · Major Revisions

English editing of the manuscript by a language expert is highly recommended. Two PDF files are attached for reviewers' comments.

Reviewer 1 ·

Basic reporting

1. Unambiguous, professional English language used throughout.
2. Abstract: Minor updates needed
a. Methods: Add gist of study design details such as division of groups and treatment assignment etc.
b. Results: Consider adding brief statement about “clinical symptoms and mortality”
3. Introduction is adequate covering the area of research with recent references.
4. Material and Methods: Please see structure changes details
• Change sub-section 2.1 as Animals, also make the details of ethical committee at the end of paragraph as starting part.
• 2.2 should be Materials
• Consider adding new separate sub-section 2.3- Experimental design: explain the details
• 2.4 LORR measurement; adjust numbering of next sub-sections accordingly.
5. The results are well summarized.
6. The references used are Adequate - reference number 31 please add details of volume, issue and page numbers of the article.
7. Figures: Minor updates to description
• Fig 2: Consider adding statement of "No statistically significant difference in recovery time among the drugs.
• Fig 3: In the description, remove “for loss of the righting reflex.” Add statement of "Etomidate and ET-26HCl had a milder BP inhibition effect than propofol."
• Fig 4: In the description, remove “for loss of the righting reflex.” Add statement of “No significant differences were observed between control group and ET-26HCl group at 15, 30 and 60 min after ACTH stimulation.”
8. Table 1: changes required, please see bolded details
• Change heading from “Number of apneas” to “Number of aged rats with apnea.”
• Include unit of time (S) for Duration of apnea.
• In the description consider adding statement of "Apnea in all rats was observed in Propofol group; no apnea in Etomidate and ET-26HCl groups."

Experimental design

1. Interesting piece of work which would add to the evidence base in this field.
2. Research is substantial, and the results would contribute to the future development of clinical studies
3. Good information regarding methodology to replicate the study.

Validity of the findings

1. Manuscript sound scientific, along appropriate use of software to analyze the data.
2. Good sample size and appropriate statistical tests are employed for this analysis.
3. The coverage of the discussion is adequate. The conclusion is appropriate including limitations.

Additional comments

This article represents the importance and need of pre-clinical studies in aged rats to establish efficacy and safety of ET-26HCl. The authors supported their hypothesis/recommendation of clinical use ET-26 HCl in elderly population clearly.

To this end, my decision is approval of the manuscript with minor revisions.

Annotated reviews are not available for download in order to protect the identity of reviewers who chose to remain anonymous.

Reviewer 2 ·

Basic reporting

Attached

Experimental design

Attached

Validity of the findings

Attached

Annotated reviews are not available for download in order to protect the identity of reviewers who chose to remain anonymous.

Reviewer 3 ·

Basic reporting

-Some ambiguity in terms of grammar
For example: Line 55, "similar cardiovascular stability and slight respiratory depression as etomidate, but mild adrenocortical inhibition" can be "was developed as a novel intravenous anesthetic with not lonly similar extent of cardiovascular stability and respiratory depression comparing to etomidate, but also milder adrenocortical inhibition."

eg2. Line 198 "Thus, the rats in this study can be translated to predict the potential advantages
199 of ET-26HCl in the elderly population" can be "Therefore, results of the current study can be translated to predict..."

-sufficient literature references

Experimental design

convincing data and reasonable experimental design in general

Validity of the findings

-mean and SD could be included in Figure 2

Additional comments

-please discuss briefly mechanisms and differences between etomidate and ET-26 hydrochloride
-limitations also include minimal evidence and calculations on EC50 and ED50

---

## Round 0.2 · accepted · Accept

The paper can be accepted for publication in its current form.